# ARTI (Adaptive Radio Tomographic Imaging): One New Adaptive Elliptical Weighting Model Combining with Tracking Estimates

**DOI:** 10.3390/s19051034

**Published:** 2019-02-28

**Authors:** Chunhua Zhu, Jiaojiao Wang, Yue Chen

**Affiliations:** College of Information Science and Engineering, Henan University of Technology, Zhengzhou 450001, China; wangjiaojiao@stu.haut.edu.cn (J.W.); 201692310@stu.haut.edu.cn (Y.C.)

**Keywords:** adaptive radio tomographic imaging, Savitzky-Golay filtering, elliptical weighting model, target tracking

## Abstract

Imaging and tracking performance suffers from the mismatch between the model and the measurements in an adaptive radio tomographic imaging system. In this paper, a model-based approach is reviewed and a new adaptive elliptical weighting model is proposed, in which the coverage of ellipse and the voxels weightings can adaptively match the actual environments, and the Savitzky–Golay smoothing filter is presented to eliminate the influence of measurement noise and multipath interference. In our proposed model, the optimal coverage of ellipse and weightings can be obtained from voxel weightings distribution inside the ellipse and pseudo-position area and trailing phenomenon. Finally, the development efforts are evaluated and validated with real experiments conducted in indoor environments for a moving target. The results have shown that the proposed algorithm can improve the accuracy of image and location estimates compared with the normalized weight model and the const-eccentricity weight model.

## 1. Introduction

With the development of wireless sensor networks (WSNs), passive target detection technology has become a hot issue in the field of positioning. In recent years, radio tomography imaging (RTI) technology based on WSNs has been developed, in which the received signal strength (RSS) measurements among the static wireless sensors is used to image the changes in the radio propagation environment in the areas of the sensors. As a result, the changes caused by the human bodies can be detected and tracked in indoor and outdoor environments with lower power and less cost [1,2]. In an RTI system, accurate localization depends on an accurate model for RSS measurements, and the elliptical model is a typical one to model RSS and to perform localization. The classical weighting model was first proposed by Joey Wilson and Neal Patwari [2,3]. However, there are some limitations for this model, such as the voxel weightings inside the ellipse being the same, which is not consistent with the actual environment. In 2014, Benjamin R. Hamilton proposed an inverse area elliptical model [4]. In this model, the voxel weightings near the LOS (line of sight) path are larger because of the shorter signal propagation path, and vice versa. To embody this characteristic, each voxel weighting is equal to the inverse of the area of the smallest ellipse containing the transceiver and this voxel. This inverse area elliptical model owns greater accuracy than others because the contribution of each voxel in one ellipse can be distinguished. However, the computational complexity is too high to locate in real time. In 2015, a const-eccentricity elliptical model is proposed in [5], which is suitable for the small coverage of ellipse with shorter distance. Therefore, the images can be reconstructed with fewer voxels, greatly reducing the noise interference. However, how to decide the weightings inside the ellipse is not considered. In [6], a geometry-based elliptical model is proposed. Here, one ellipse is divided into several different areas with two kinds of weightings by the line-of-sight and non-line-of-sight, and the orthogonal matching pursuit (OMP) algorithm is combined to eliminate extra bright spots in image reconstruction. To this end, the accuracy of positioning is up to 23.8% for one target over the classical weighting model [2,3]. However, the weightings representing the obstacle to communicate in line-of-sight path and non-line-of-sight path need be decided by empirical experiment, which is unsuitable in the real-time positioning.

Reviewing the existing research, considering the real-time capability and the actual environments matching, the new elliptical model should be developed to match the measurements and eliminate the influence of measurement noise and multipath interference [7,8]. Based on this, in this paper, we propose a new model, which includes the adjustment of the coverage of ellipse and the selection of voxel weightings. Specifically, the coverage of ellipse will be adjusted by an adaptive parameter φ, and the voxel weightings will be changed with the distance *h* between the voxel and the LOS path, where the distance attenuation factor $e^{-h}$ is introduced to define the weighting values. The proposed ellipse model can adaptively adjust the voxel weightings according to the position of each voxel and the corresponding signal propagation, which is consistent with the actual environments. Besides this, the Savitzky–Golay smoothing filter [9,10,11,12] is presented to improve the location estimates as Figure 1. The Savitzky–Golay algorithm was first proposed by Savitzky and Golay in 1964, which can smooth the data stream by local polynomial least squares fitting in the time domain. As one special kind of low-pass filter, the Savitzky–Golay filter can eliminate the residual measurement noise and keep the shape and width of the signal unchanged. More importantly, for the Savitzky–Golay algorithm, the cost in computation is lower and there is no excessive requirement for the computer memory and data processing ability. 

This paper is organized as follows: in Section 2, the principle of RTI and the new adaptive elliptic model are introduced. The images for moving target are reconstructed and the positioning performance is analyzed in Section 3. Finally, the conclusion is presented in Section 4.

## 2. Adaptive Radio Tomography Imaging

### 2.1. Introduction of RTI

As shown in Figure 2, there are *L* sensor nodes deployed around the monitor area. A line of sight (LOS) path is established between any two sensor nodes, and the total number of the links can be denoted as *M* = *L*(*L* − 1)/2. The monitor area is divided into *N* voxels. When a target enters the monitor area, the RSS value will be changed as [1,13].
(1)y=Wx+n
where *y* denotes the changes of all the links inside the network, *W* means the shadow weight matrix, *x* is the signal fading value in the voxel, *n* represents the noise. To estimate an image from the measured data, we need to find the optimal solution with the LS (least square)error:(2)$$x_{LS}=arg_{x}\min\;{\left\|Wx-y\right\|}_{2}^{2}$$

The LS solution can be obtained by setting the gradient zero:(3)$$x_{LS}={\left(W^{T}W\right)}^{-1}W^{T}y$$

In Tikhonov regularization, the energy term is added to the LS formula, and the objective function is obtained [13]:(4)$$f\left(x\right)=\frac{1}{2}{\left\|Wx-y\right\|}^{2}+\alpha {\left\|Qx\right\|}^{2}$$
where *Q* is a Tikhonov regularization matrix [14].

The regularization of the image should include components in the vertical and horizontal directions. The matrix $D_{X}$ is a horizontal difference operator, and $D_{Y}$ is a vertical difference operator. The regularization functions can be written as follows:(5)$$f\left(x\right)=\frac{1}{2}{\left\|Wx-y\right\|}^{2}+\alpha \left({\left\|D_{X}x\right\|}^{2}+{\left\|D_{Y}x\right\|}^{2}\right)$$

Then, we can obtain the derivative of *f*(*x*) and set it as zero, and the results can be calculated as:(6)$$x={\left(W^{T}W+\alpha \left(D_{X}^{T}D_{X}+D_{Y}^{T}D_{Y}\right)\right)}^{-1}W^{T}y$$

### 2.2. Adaptive Elliptical Weight Model

A weight model is used to decide whether the voxel contributes to image reconstruction. In general, the RTI technology can be depicted as in Figure 3a,b when it is used in the elliptical weight model, where the foci of ellipse are the corresponding two sensor nodes inside this ellipse. In the classical weighting model [2,3] (referred to as Model 1), the coverage of ellipse is one constant and the weighting of each voxel inside one ellipse is set as 1; if the voxel is outside this ellipse, the weighting is set as 0, which is described as
(7)$$W_{i,j}=\frac{1}{\sqrt{d_{i}}}\{\begin{array}{cc} 1 & d_{ij}(1)+d_{ij}(2)<\lambda \\ 0 & otherwise \end{array}$$
where $d_{i}$ is the length of link *i*, and $d_{ij}(1),d_{ij}(2)$ are the distances between voxel j and two sensor nodes, respectively. λ is an adjustable parameter to determine the range of the ellipse, which is a constant decided by the experiments. Because the distances of the links between any two sensor nodes are different, the weightings of voxels inside the ellipse need to be multiplied by the square root of the distance of two nodes, such as link1 and link2 in Figure 3b.

According to Equation (7), for different $d_{i}$, the corresponding coverage of ellipse will be a constant because of the fixed λ. In this case, the weightings inside the ellipse will also be a constant, which results that more measurement noise is introduced for the shorter links, and the quality of reconstructed images is degraded.

In [5], the const-eccentricity weight model (referred to as Model 2) is presented to solve this problem, which can be described as:(8)$$W_{i,j}=\frac{1}{\sqrt{d_{i}}}\{\begin{array}{cc} 1 & d_{ij}(1)+d_{ij}(2)<d_{i}/\varepsilon \\ 0 & otherwise \end{array}$$
where *ε* is centrifugal rate of ellipse, which is described as:$$\varepsilon =\frac{d_{\max}}{d_{\max}+\lambda }$$
where $d_{\max}$ is the length of the longest link in the RTI network.

In the const-eccentricity weight model, the coverage of ellipse will change proportionally with link distance, improving the positioning performance. However, for Models 1 and 2, the attenuation contribution of different voxels in the same ellipse cannot be distinguished. In fact, a closer distance between the voxel and LOS path results in a larger attenuation contribution of this voxel. Therefore, the weightings of the voxels in one ellipse should be distinguished according to the distance between these voxels and LOS path. Considering the distance attenuation characteristics of wireless signal, a distance attenuation factor $e^{-h}$ is introduced in the elliptical weight model, which is described as
(9)$$W_{i,j}=\{\begin{array}{cc} e^{-h} & d_{ij}(1)+d_{ij}(2)<d_{i}+\varphi \lambda \\ 0 & otherwise \end{array}$$
where *e* is the base of the natural logarithm and *h* denotes the distance between each voxel inside the ellipse and the LOS path, and *φ* is an adaptive parameter that can adjust the coverage of the ellipse. Mathematically, *φ* is described as:(10)$$\varphi =\frac{d_{\max}}{d_{i}}$$

From Equation (9), a shorter link distance leads to the larger φλ and a bigger coverage of ellipse. Thus, the variation characteristic is inverse with Model 2. This is because the attenuation contribution of different voxels in the same ellipse can be distinguished. Therefore, the larger coverage of ellipse is, and the more improvement will be brought to the quality of reconstruct images. To this end, our proposed ellipse weight model is adaptive to the actual environments, which is called Model 3. As shown in Figure 4, for a same link in the network, it can be seen clearly that the proposed adaptive ellipse weight model has a larger coverage of ellipse in comparison with Model 1 and Model 2. Besides this, the voxel weightings distribution also exhibits a significant difference, as depicted in Figure 5.

From Figure 5, in Models 1 and 2, the weightings of voxels inside the ellipse are constant. However, in Model 3, the closer distance between the voxels and LOS path resulted in larger weightings. Therefore, the selection of weightings was more consistent with the real environment. Besides this, for the same link distance, the coverage of ellipse was expanded, and more voxels in one ellipse were used to reconstruct images.

### 2.3. Positioning Estimation

According to the optimal weightings and target location based on LS and Tikhonov regularization, the estimated target position with the greatest attenuation is recorded as ($x_{i}$, $y_{i}$) in the coordinate axis, *I* = 1,2,…,72, which is the sampled position index.

To further eliminate the residual noise or interference, the Savitzky–Golay Filter was adapted for its advantage of lower complexity in computation and real-time performance [9]. The positioning data achieved by Equation (6) can be fitted by a LS-based polynomial to obtain one new set of data. Due to the need of windowing the moving process in the Savitzky–Golay filter, the windowed data was recorded symmetrically as $x_{i},\;i=-m,\cdots ,0,\cdots ,m$, that is, the window width is *n* = 2*m* + 1, *m* = 0. 1,2,…,35 for the given 72 samples. The (*k* − 1)-order fitting polynomial is
(11)$$y=\sum_{l=0}^{k-1}a_{l}x^{l}=a_{0}+a_{1}x+a_{2}x^{2}+\cdots +a_{k-1}x^{k-1}$$
here, *y* is one new set of data. From Equation (11), there are 2*m* + 1 equations and a linear equation with *k* unknown numbers. Only under the condition *n* > *k* can Equation (11) be solved. Recording the fitting error as $e_{i},\;i=-m,\cdots ,0,\cdots ,m$, Equation (11) can be rewritten as *k*-order equations,
(12)$$\left(\begin{array}{c} y_{-m}\\ y_{-m-1}\\ \vdots \\ y_{m} \end{array}\right)=\left(\begin{array}{cccc} \begin{array}{c} 1\\ 1\\ \vdots \\ 1 \end{array} & \begin{array}{c} x_{-m}\\ x_{-m+1}\\ \vdots \\ x_{m} \end{array} & \begin{array}{c} \cdots \\ \cdots \\ \vdots \\ \cdots \end{array} & \begin{array}{c} x_{-m}{}^{k-1}\\ x_{-m+1}{}^{k-1}\\ \vdots \\ x_{m}{}^{k-1} \end{array} \end{array}\right)\left(\begin{array}{c} a_{0}\\ a_{1}\\ \vdots \\ a_{k-1} \end{array}\right)+\left(\begin{array}{c} e_{-m}\\ e_{-m+1}\\ \vdots \\ e_{m} \end{array}\right)$$
where $a_{0},a_{2},a_{3},\cdots ,a_{k-1}$ are the fitting parameters, which can be solved by LS method. For the given fitting parameters $a_{il}$, the least fitting mean square error (MSE) is defined as
(13)$$E=\sum_{i=-m}^{m}e_{i}^{2}=\sum_{i=-m}^{m}{\left[y_{i}-x_{i}\right]}^{2}=\sum_{i=-m}^{m}\left[\sum_{l=0}^{k-1}a_{il}[x_{i}{}^{l}-x_{i}{]}^{2}\right]$$
let the first derivative of the function *E* equals to zero, that is
(14)$$\frac{\partial E}{\partial a_{ir}}=0,\;r=0,1,2,\cdots ,k-1$$
thereby, the minimum of MSE (mean square error)can be obtained, which corresponds to the optimal fitting parameters in Equation (12).

Representing the Equation (12) as matrix form,
(15)$$Y_{(2m+1)\times 1}=X_{(2m+1)\times k}\cdot A_{k\times 1}+E_{(2m+1)\times 1}$$
here, $A=[a_{0},a_{2},a_{3},\cdots ,a_{k-1}]$. By LS method, the optimal solution is $\hat{A}$ as
(16)$$\hat{A}={\left(X^{T}\cdot X\right)}^{-1}\cdot X^{T}\cdot Y$$

Then, the filtering value $\hat{Y}$ for Y is
(17)$$\hat{Y}=X\cdot A=X\cdot {\left(X^{T}\cdot X\right)}^{-1}\cdot X^{T}\cdot Y=B\cdot Y$$
where $B=X\cdot {\left(X^{T}\cdot X\right)}^{-1}\cdot X^{T}$.

In the above filtering process, the window width is *n* = 2*m* + 1, which must satisfy the conditions of *n* > *k* and ranges from 0 to 35. There exists one optimal window width corresponding to the lowest positioning mean square error (MSE), which can be analyzed in the following experiment in detail, because the optimal window width can be different for different experiment environments.

## 3. Experiment Results

### 3.1. Experiment Design

The experiment was designed at Beijing Institute of Technology, and the network was placed in an office where there were 16 TI CC2530 (Texas Instruments Inc, Texas, USA) sensor nodes within the network. In order to reduce the influence of ground reflection, the nodes were placed on the bracket, and the height from the ground is 1 m. At the same time, in order to suppress the reflection signals from the walls, ceilings, and other external monitoring areas, a flat directional antenna with horizontal beam width of 110 degrees and vertical beam width of 30 degrees were used for this paper. The node fully supports the IEEE 802.15.4 protocol, and the maximum transmission power is 4.5 dBm, which is enough to cover the whole monitoring area. In order to measure RSS of all links quickly, this paper adopted a token-ring-like communication protocol developed by Wilson [1]. In this protocol, in order to avoid collision, each node was assigned a unique ID number as identification and the sending order depended on the ID number of the node. When all nodes completed a signal transmission, RSS of all links were updated and Figure 6 presents the experimental environment. The monitoring area measured 6 m × 6 m, the voxel interval was set to 0.1 m × 0.1 m, and the number of voxels was 3600. The parameter λ is 0.05 in Equation (9). One person could move in the monitoring area at a constant velocity to obtain the target trajectory.

### 3.2. Tomographic Reconstruction

According to the coordinate position of each node and Equation (9), we calculated the weight matrix *W* of dimension 240 × 3600, and obtained the RSS attenuation matrix ***y*** of dimension 240 × 1 by the received signal strength changes for one target and no target. The voxel attenuation matrix ***x*** was calculated using Equation (6). For the given three target positions, the reconstructed images for Model 1, Model 2, and Model 3 are shown in Figure 7, respectively, of which the dark spot represents the target position. From Figure 7, it can be seen that the bright spot is more concentrated and the pseudo-position area inside the reconstructed image of Model 3 shrinks obviously.

In the reconstructed image, there exists the pseudo-positions and trailing phenomenon because of residual noise. By selecting the coverage of ellipse and voxel weightings adaptively, the proposed model can decrease measurement noise and match the measurement environment. For three different target positions, the corresponding number of relative serious attenuation voxels, and the number of pseudo-position voxels are shown in Figure 8, respectively. From Figure 8, it can be seen that the number of pseudo-position voxels and the number of serious attenuation are decreased obviously in Model 3, compared with Model 1 and Model 2.

### 3.3. Filtering Parameter Selection

From Section 3.2, the target position with the greatest attenuation was recorded as ($x_{t}$, $y_{t}$) in the coordinate axis. In order to improve the positioning performance, the estimated position data set was inputted into the Savitzky–Golay filter. From Section 2.3, in the above filter process, the window width is *n* = 2*m* + 1, and when *n* > *k*, the equation set can be solved, so the selection of parameter *n* affects the filtering performance. In this paper, the optimal window width was decided by empirical experiment.

The average locating error of the Savitzky–Golay filter in different window widths is shown in Figure 9. For different window width from 3 to 21, there is a dedicated average locating error. Thus, in this paper, the estimated target trajectory was filtered under the optimal window.

### 3.4. Tracking Performance Analysis

For a moving target, the target trajectory under different approaches is shown in Figure 10. The green curve represents the real target trajectory, the blue curve is the target trajectory estimated by Equation (6), and the red curve is obtained by the proposed ellipse model combining the Savitzky–Golay filter with the optimal window width. It can be seen clearly that the trajectory obtained by the proposed model is closer to the real trajectory. For the three selected target positions, the MSE of the proposed approach is compared with the existing approaches, as shown in Figure 11. The average accuracy of the three-positioning estimation is up to 67.5% over the classical weighting model, and 54.7% over the const-eccentricity elliptical model.

## 4. Conclusions

In this paper, we proposed an adaptive elliptical weighting model and adopted the Savitzky–Golay filter for image reconstruction to enhance the positioning accuracy in an RTI system. The new model concerned the inside of the ellipse, in which the adaptive factor *φ* and the attenuation factor $e^{-h}$ were adopted to zoom the coverage of elliptical and coordinate voxels weightings to distinguish the differece of path loss in the target area, respectively. Therefore, the proposed model can better match the actual measurement environments. Besides this, the Savitzky–Golay filtering eliminated the residual noise and interference in target trajectory, and achieved better accuracy. Compared with the existing algorithms, the proposed model-based approach can resist the measurement noise and multipath interference, match the actual environments, and realize real-time positioning with lower complexity.

## Figures and Tables

**Figure 1 sensors-19-01034-f001:**
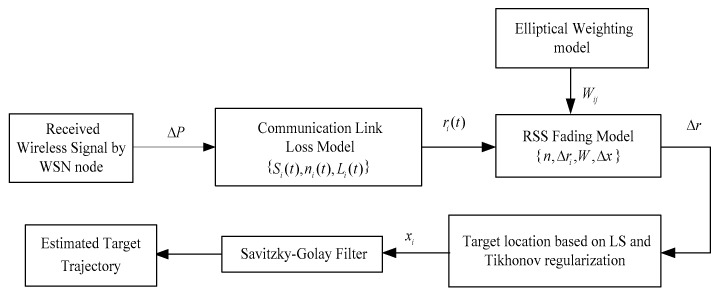
Radio tomography imaging (RTI) sensor node deployment.

**Figure 2 sensors-19-01034-f002:**
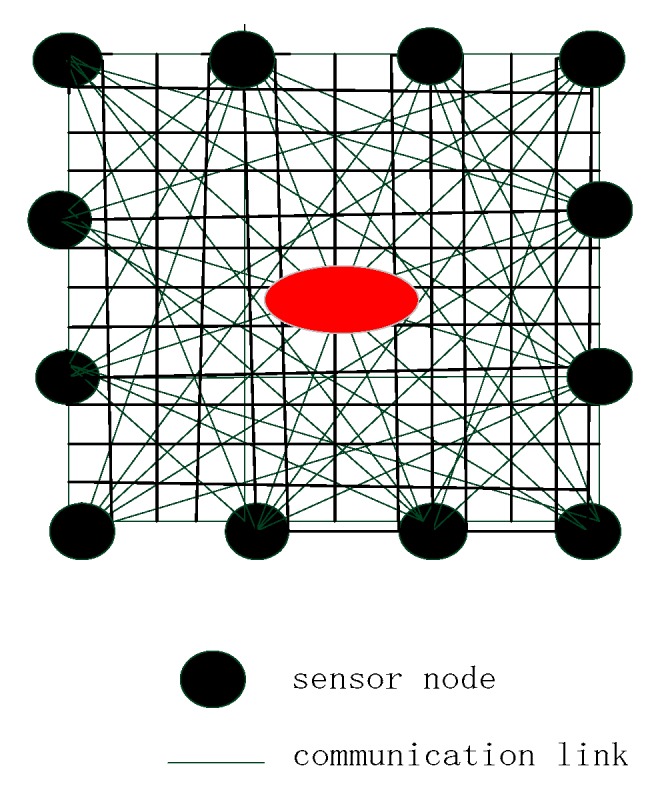
RTI sensor node deployment.

**Figure 3 sensors-19-01034-f003:**
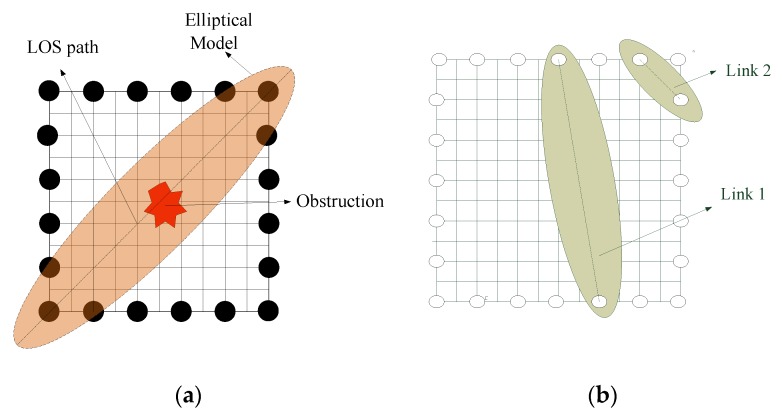
The elliptical weight model. (**a**) The longest link; (**b**) different links.

**Figure 4 sensors-19-01034-f004:**
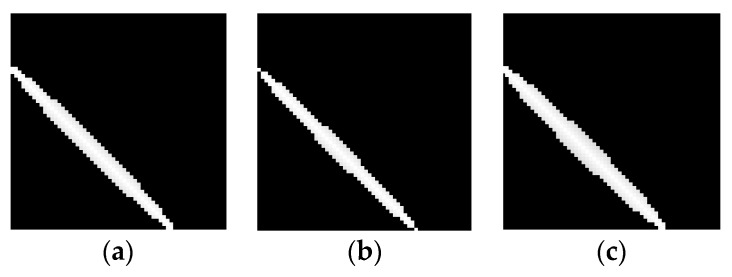
The coverage of different ellipse weight models (**a**) Model 1; (**b**) Model 2; (**c**) Model 3.

**Figure 5 sensors-19-01034-f005:**
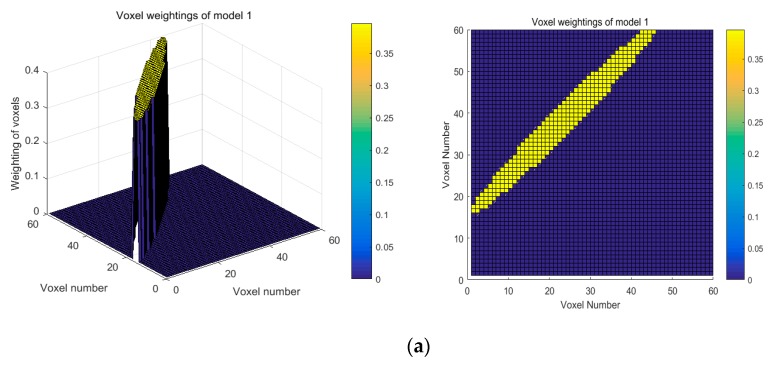
Voxel weightings distribution of different models (**a**) Model 1; (**b**) Model 2; (**c**) Model 3.

**Figure 6 sensors-19-01034-f006:**
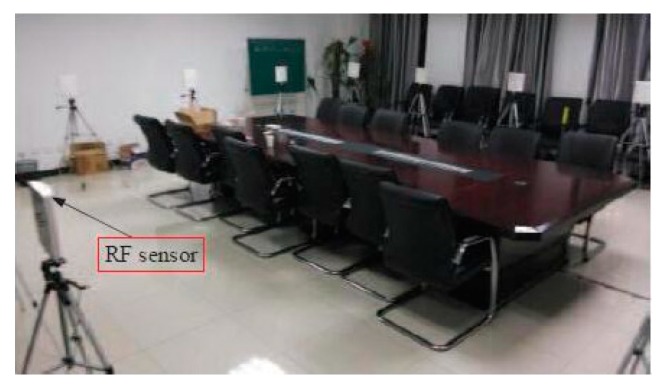
RTI sensor node deployment.

**Figure 7 sensors-19-01034-f007:**
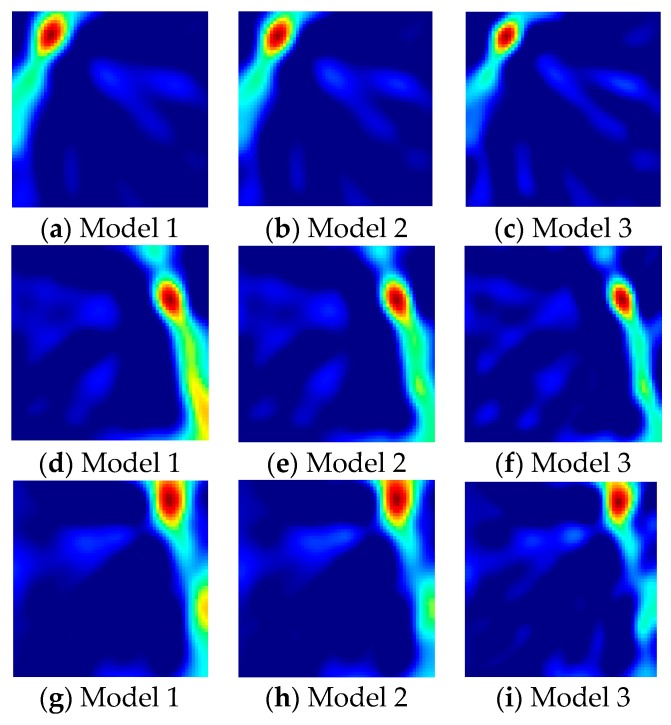
Reconstructed images by Three Models (**a**–**i**).

**Figure 8 sensors-19-01034-f008:**
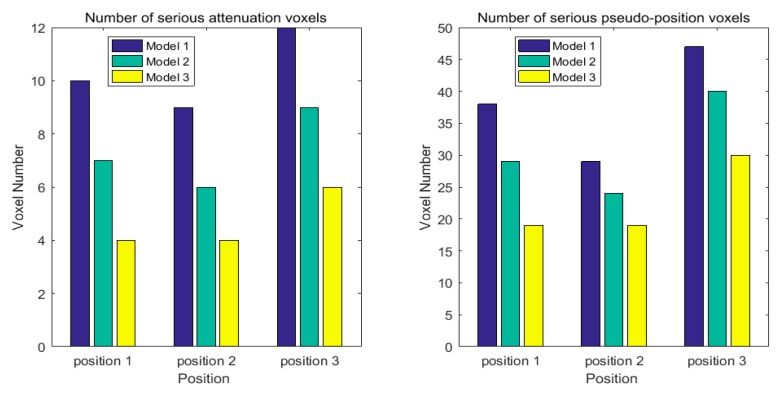
Positioning performance from reconstructed images.

**Figure 9 sensors-19-01034-f009:**
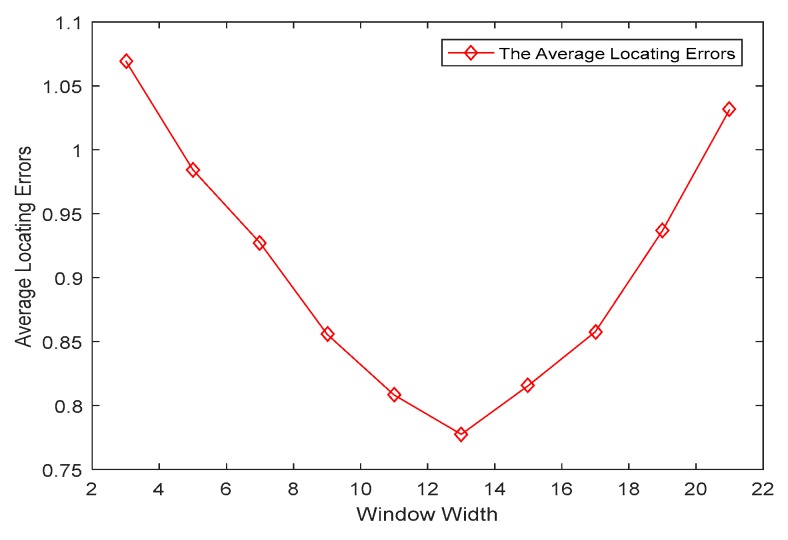
The average location error curve.

**Figure 10 sensors-19-01034-f010:**
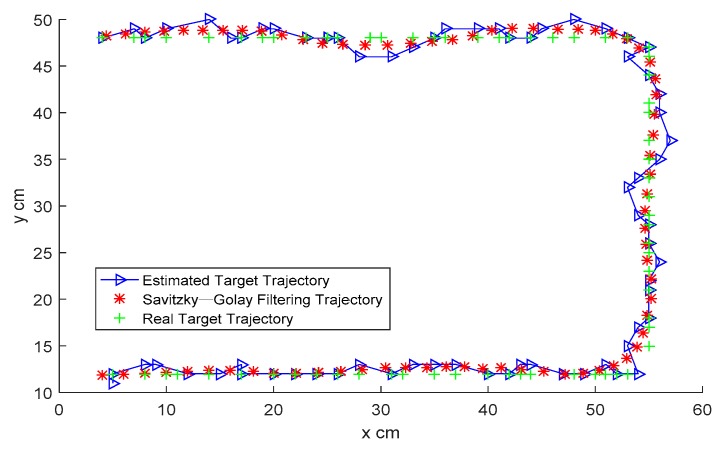
Moving target trajectory.

**Figure 11 sensors-19-01034-f011:**
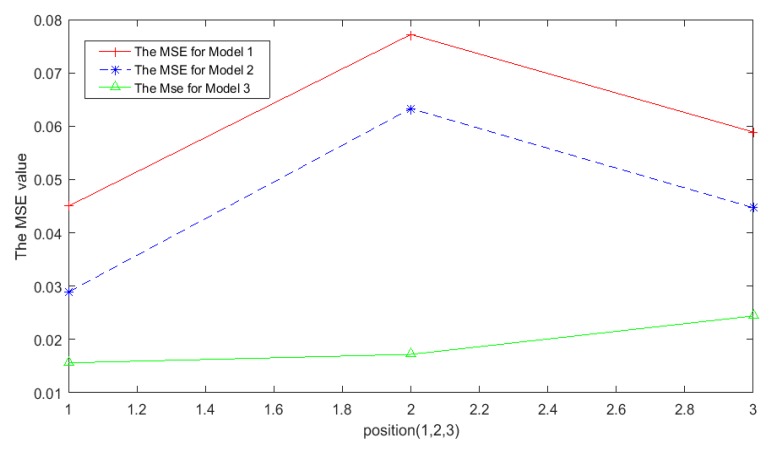
The positioning error of different approaches.

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
