# Peer review of "ARTI (Adaptive Radio Tomographic Imaging): One New Adaptive Elliptical Weighting Model Combining with Tracking Estimates"

_sensors, 2019, doi:10.3390/s19051034_

Round 1

Reviewer 1 Report

The manuscript presents the imaging and tracking performance based on an improved radio tomography imaging approach. The results are compared with other two available imaging models and the results demonstrated the improvement. The research makes a useful contribution to the literature and the contents interested to the reader of the sensor journal. However, there are major/minor issues related with the contents which need to be addressed before the manuscript accepted to publication.

 (1). Part of the parameters are missing between lines 99 and 107 and it is difficult to following the equation of (7). The paper need to be improved as there exist typos to make it briefer and easier to follow.

(2). The weighting coefficient “w” should be keep the same among equations (7), (8) and (9).  In equation (8), it is a capital “W”.

(3). In line 195, the parameter of “λ” is set as 0.05, please give justification for this value.

(4).  For the RTI sensor as dispatched in Fig.6, the sensor is 3D dimension and the fringe effect from the bottom and top must be considered in the image processing. How to consider the fringe effect and how reduce the noise effect from bottom and top effect? The review would like the authors consider these issues in the revise version.

Author Response

Dear Editors and Reviewers:

        Thank you for your letter and for the reviewers’ comments concerning our manuscript entitled “ARTI: One New Adaptive Elliptical Weighting Model Combining with Tracking Estimates”. Those comments are all valuable and very helpful for revising and improving our paper, as well as the important guiding significance to our researches. We have studied comments carefully and have made correction which we hope meet with approval. Revised portion are marked in red in the paper. The main corrections in the paper and the responds to the reviewer’s comments are as following:

Response to the first reviewer’s comments:

Point 1: Part of the parameters are missing between lines 99 and 107 and it is difficult to following the equation of (7). The paper need to be improved as there exist typos to make it briefer and easier to follow.

Response 1: We are very sorry for our negligence for parameter missing. We have made correction in our revision.

Point 2: The weighting coefficient “w” should be keep the same among equations (7), (8) and (9).  In equation (8), it is a capital “W”.

Response 2: We are very sorry for our incorrect writing and we have made correction in our revision.

Point 3:  In line 195, the parameter of “λ” is set as 0.05, please give justification for this value.

Response 3: When  λ is very small, the elliptical model can be degenerated into a straight line, which is considered that only the target of the line of sight path has an effect on the link signal attenuation. When  λ  increases gradually, the short axis of the ellipse becomes longer and the area of the ellipse becomes larger, the influence of energy loss of multipath signal can be taken into account. Because the energy of the link signal mainly concentrates on the LOS path, the large-scale fading (shadow fading) of the link can not be extracted well when λ is too large, and because the link signal is interfered by multi-path, when λ is too small, the assumed signal energy is too concentrated, which will also cause distortion of the imaging results. Therefore, it is necessary to select a suitable control parameter. In this paper, the most suitable elliptical parameter is selected as  λ=0.05  by analysis and experiment.

Point 4: For the RTI sensor as dispatched in Fig.6, the sensor is 3D dimension and the fringe effect from the bottom and top must be considered in the image processing. How to consider the fringe effect and how reduce the noise effect from bottom and top effect? The review would like the authors consider these issues in the revise version.

Response 4: The covariance matrix in imaging space represents the priori relationship among the pixels in an unknown image. The small attenuation factor will cause the imaging object to be discrete which can’t play a priori constraint role; the large attenuation factor will cause the loss of edge information of the imaging object and reduce the imaging accuracy. We can improve the edge effect and reduce the noise interference by choosing the appropriate attenuation factor in the spatial covariance matrix.

We tried our best to improve the manuscript and made some changes in the manuscript. We appreciate for Reviewers’ warm work earnestly, and hope that the correction will meet with approval.

Once again, thank you very much for your comments and suggestions. 

Reviewer 2 Report

If I understand it correctly, the paper describes a new data processing method in RTI (radio tomography imaging) where non-uniform, separation distance related, weighting is used to increase contrast of reconstructed image, and trajectory tracking is smoothed by a math process (Savitzky-Golay smoothing). A case study is presented to demonstrate improvements this new method brings compared to some existing ones. I have the following general questions that I think if adequately addressed should improve the quality (especially the rigour) of the paper.

1. The Conclusions section reads like a summary, not like conclusions. The proposed new method is meant to be an improvement over the existing ones, thus quantitative measure of such improvement is expected in the conclusions. But with just one (and simple) test case, it probably wouldn’t be that convincing anyway.

2. What appears in Fig.4 contradicts with the narrative in the main text above the figure. The claim is Model 3 has the largest coverage, but the figure appears to showing Model 1 is the case.

3. Line 196: “constant”. Why constant speed is important, and how important it is? What if someone moves at variable speed?

4. Line 232: “empirical”. How does this work in practice? What affect the error level? To quantify the “error”, you will need to know what the correct answer is; otherwise, it’s called “uncertainty”. In reality, if the user knows the answer, why would they bother to use your method which needs calibration in each case to work?

5. Figure 9: the error range is only 0.3 percentage point. It doesn’t seem a ‘big deal’. Comment on how significant this is.

6. Figure 10: Savitzky-Golay is a math process, it will smooth everything you throw at it, regardless of physical reality. What’s the risk of this smoothing out features (corners or sharp turns) that are actually there, or how do you know you’re not doing that? What if your tracer follows a zigzag path with variable speed?

7. What if you have multiple and moving objects inside your monitoring zone? More test cases need to be presented I think.

Minor points:

8. Space between word and bracket. Example, “…networks(WSNs)” should be “…networks (WSNs)”.  

9. “foci…is”: grammar.

10. Missing symbols in the two paragraphs below Equation (7).

11. Line 132: “more conductive to reconstruct images” – meaning what?

12. Figure 7 caption: “image” should be “images”.

13. Line 234: “dedicated average”? Meaning what?

Author Response

Dear Editors and Reviewers:

        Thank you for your letter and for the reviewers’ comments concerning our manuscript entitled “ARTI: One New Adaptive Elliptical Weighting Model Combining with Tracking Estimates”. Those comments are all valuable and very helpful for revising and improving our paper, as well as the important guiding significance to our researches. We have studied comments carefully and have made the corresponding correction, and the revised has been marked in red. The detailed corrections and the response to the reviewer’s comments are as following:

Response to the second reviewer’s comments:

Point 1: The Conclusions section reads like a summary, not like conclusions. The proposed new method is meant to be an improvement over the existing ones, thus quantitative measure of such improvement is expected in the conclusions. But with just one (and simple) test case, it probably wouldn’t be that convincing anyway.

Response 1: It is really true as Reviewer suggested that only one test case is presented.  In this paper, the experiment in one office is performed and chose  typically because it turns out that the positioning performance  will be more complicated for indoor environment compared with outdoor one, and the indoor experiment generally is used to evaluate the positioning error as in the majority of references[1, 2,3,4  ] in our manuscript.

1.     Liu H , Liu C , Shu W , et al. Radio-Frequency Tomographic Tracking of a Time-Varying Number of Targets with Wireless Sensor Networks[J]. 2013.

2.     Hong Yun-tao, Wu Chun-xue, MIAO Hong-min. Object tracking using radio tomography imaging and Kalman filter [J]. Information Technology, 2015(7):55-58.(In Chinese)

3.     Kaltiokallio O , Bocca M , Patwari N . Enhancing the accuracy of radio tomographic imaging using channel diversity.[C]// IEEE International Conference on Mobile Ad-hoc & Sensor Systems. IEEE Computer Society, 2012.

4.     Wei B , Varshney A , Petwari N , et al. dRTI: Directional Radio Tomographic Imaging[C]// International Conference on Information Processing in Sensor Networks. ACM, 2015.

Point 2: What appears in Fig.4 contradicts with the narrative in the main text above the figure. The claim is Model 3 has the largest coverage, but the figure appears to showing Model 1 is the case.

Response 2: The Fig.4 has been substituted and the new Fig.4 corresponds to the Model 3 in revised manuscript.

Point 3: Line 196: “constant”. Why constant speed is important, and how important it is? What if someone moves at variable speed?

Response 3: Theoretically, for the presented new positioning method, all environments should be considered to prove the effectiveness; however, it is not the facts as most of the existing studies. The main reason is that the positioning performance won’t produce more changes for the cases of variable speeds, especially for faster positioning speed.  In this paper, the average step speed of 0.4 meter per second is adopted to evaluate the presented algorithm, and. Nonetheless, the tracking performance of the proposed RTI method will have some difference for various motion states and positions, which should be researched in detail, and the maximum of variable speed and side effects can be considered in the future research.

Point 4:  Line 232: “empirical”. How does this work in practice? What affect the error level? To quantify the “error”, you will need to know what the correct answer is; otherwise, it’s called “uncertainty”. In reality, if the user knows the answer, why would they bother to use your method which needs calibration in each case to work?

Response 4:  Here, “empirical” isn’t be used properly, in fact, the optimal window width can be different for different experiment environments. For each given window width and experiment environment, the optimal window width can be obtained by searching the minimum of the final positioning error. Therefore, in practice, the proposed algorithm can adapt to the other works.

Point 5:  Figure 9: the error range is only 0.3 percentage point. It doesn’t seem a ‘big deal’. Comment on how significant this is.

Response 5: In Figure 9, the improvement of 0.3 percentage point is only for the target positions from the Savitzky-Golay filtering, not for the whole reconstructed images. Therefore, for the whole reconstructed images, the target positioning error will be turn lower, and the significant improvement will be achieved as Figure 11.

Point 6: Figure 10: Savitzky-Golay is a math process, it will smooth everything you throw at it, regardless of physical reality. What’s the risk of this smoothing out features (corners or sharp turns) that are actually there, or how do you know you’re not doing that? What if your tracer follows a zigzag path with variable speed?

Response 6:  In our works, the Savitzky-Golay filtering is performed only for the obtained positions from the proposed elliptical model as Equ. (6), not for the whole reconstructed images. Moreover, for the complicated or sensitive environment, such as corners or sharp turns, the samples of positioning trajectory can be increased to solve the above problem.

Point 7: What if you have multiple and moving objects inside your monitoring zone? More test cases need to be presented I think.

Response 7: In fact, the influence of different targets on RSS is mutual coupling, so it is necessary to take into account the interaction between targets and establish a more accurate RSS observation model for multi-targets. It is an urgent problem that RTI needs to overcome in application.

 Point 8: Space between word and bracket. Example, “…networks(WSNs)” should be “…networks (WSNs)”. 

Response 8: The space between word and bracket and the other writing mistakes have been checked all through the manuscript and corrected in the revised paper.

 Point 9: “foci…is”: grammar.  

Response 9:  The pointed grammar error has been revised by replacing “is” for “are”, besides, we have gone over the whole paper and revised the other ones.

Point 10:  Missing symbols in the two paragraphs below Equation (7) .

Response 10: We are very sorry for our negligence. The missing symbols in the two paragraphs below Equation (7) have been increased in the revised manuscript.

Point 11: Line 132: “more conductive to reconstruct images” – meaning what?  

Response 11:

For the shorter link, the more pixels can be obtained in the presented elliptical model (model 3), compared with the other models, thereby the quality of the reconstructed images can be improved greatly because of the increased pixel information. Precisely, the sensence ““the larger coverage of ellipse will be more conductive to reconstruct images.””  should be revised for “the larger coverage of ellipse is, and the more improvement will be brought to the quality of reconstruct images.”

Point 12:  Figure 7 caption: “image” should be “images”.

Response 12: In Figure 7, “image” has been revised for “images”.

Point 13:   Line 234: “dedicated average”? Meaning what?

Response 13: This word “dedicated” should be deleted. We are so sorry for the ambiguity.

We have tried our best to improve the manuscript and made some changes in the manuscript. We appreciate for Reviewers’ warm work earnestly, and hope that the correction will meet with approval.

Once again, thank you very much for your comments and suggestions. 

Round 2

Reviewer 2 Report

The authors have made a few minor changes to the text, but my point remains: the work is inconclusive. On the other hand, for a report on work-in-progress, I think the current version may be acceptable.